# Mitigating Exponential Mixed Frequency Growth through Frequency Selection

## Abstract

Quantum machine learning research has expanded rapidly due to potential computational advantages over classical methods. Angle encoding has emerged as a popular choice as feature map (FM) for embedding classical data into quantum models due to its simplicity and natural generation of truncated Fourier series, providing universal function approximation capabilities. Efficient FMs within quantum circuits can exploit exponential scaling of Fourier frequencies, with multi-dimensional inputs introducing additional exponential growth through mixed-frequency terms. Despite this promising expressive capability, practical implementation faces significant challenges. Through controlled experiments with white-box target functions, we demonstrate that training failures can occur even when all relevant frequencies are theoretically accessible. We illustrate how two primary known causes lead to unsuccessful optimization: insufficient trainable parameters relative to the model's frequency content, and limitations imposed by the ansatz's dynamic lie algebra dimension, but also uncover an additional parameter burden: the necessity of controlling non-unique frequencies within the model. To address this, we propose near-zero weight initialization to suppress unnecessary duplicate frequencies. For target functions with a priori frequency knowledge, we introduce frequency selection as a practical solution that reduces parameter requirements and mitigates the exponential growth that would otherwise render problems intractable due to parameter insufficiency. Our frequency selection approach achieved near-optimal performance (median $R^2 \approx 0.95$) with 78% of the parameters needed by the best standard approach in 10 randomly chosen target functions.

## 1 Introduction

The identification of latent relationships within complex, high-dimensional datasets represents a fundamental challenge across numerous scientific and commercial domains. Drug development, for example, requires modeling quantum effects such as electron delocalization and chemical bonding, which leads to exponential scaling when computed classically (Niazi, 2025). Financial markets exhibit complex feature dependencies that traditional modeling approaches struggle to capture, as they typically assume constant correlations and neglect tail dependencies (Kolawole, 2024). Quantum machine learning (QML) emerges as a promising paradigm to address these challenges, offering theoretical advantages over classical approaches. Studies demonstrate that quantum models can represent more complex functions than classical counterparts (Mitarai et al., 2018), exhibiting higher effective dimensionality and faster training compared to traditional neural networks (Abbas et al., 2021). Substantial challenges remain for QML, however. Barren plateaus limit the number of trainable qubits (Ragone et al., 2024), while NISQ hardware constraints hinder practical deployment. Although qubit counts continue rising and error rates are improving, circuit depth must remain shallow to mitigate noise accumulation, making efficient parameter utilization critical. Analogous to activation functions in classical neural networks, quantum circuits achieve non-linearity through angle encoding, data re-uploading, and entangling gates—mechanisms that, as shown by Schuld et al. (2021), enable quantum models to represent truncated Fourier series. For successful training in supervised learning tasks, the following conditions have been established as necessary: 1) having a sufficient model frequency spectrum, together with 2) at least an equal number of parameters to control the model frequencies, and 3) enough parameters to exceed the dimension of the dynamic lie

algebra (DLA) to avoid spurious local minima (Schuld et al., 2021; Larocca et al., 2023). Through white box experiments with known target functions, we demonstrate that these conditions are not sufficient, however, as the model also contains non-unique model frequencies that need to be controlled. With exponential scaling in higher dimensions, the number of required parameters to control all frequencies quickly becomes infeasible. It is essential, therefore, to reduce the number of model frequencies to those present in the target functions. We propose including only selected frequencies in the model spectrum rather than a dense spectrum containing numerous unnecessary elements, achieving consistently superior $R^2$ scores with lower variance than the standard dense frequency approach. A second approach aids the suppression of duplicate model frequencies by initializing weight parameters close to 0 to reduce parameter requirements.

We summarize our contributions as follows:

- A systematic analysis consolidating existing parameter sufficiency conditions, revealing that additional control parameters for duplicate frequencies are necessary for function approximation with quantum circuits
- A near-zero weight initialization scheme that reduces the parameter count required for trainable quantum models
- A frequency selection algorithm that extends parameter-efficient training to higher-dimensional quantum machine learning tasks

## 2 THEORETICAL BACKGROUND

### 2.1 VARIATIONAL QUANTUM CIRCUITS

Variational quantum circuits (VQCs) represent a prominent class of QML algorithms that integrate quantum operations with classical optimization. Classical data is encoded into a quantum state $|\Psi_0(x)\rangle = S(x)|0\rangle$ via a FM $S(x)$. This encoded state is then processed by a parametrized ansatz $W(\boldsymbol{\theta})$ containing trainable parameters $\boldsymbol{\theta}$, producing the quantum state: $|\Psi(x,\boldsymbol{\theta})\rangle = W(\boldsymbol{\theta})S(x)|0\rangle$. The quantum state is measured using an observable $M$, yielding the expectation value that defines the cost function: $C(x,\boldsymbol{\theta}) = \langle\Psi(x,\boldsymbol{\theta})|M|\Psi(x,\boldsymbol{\theta})\rangle$. A classical optimizer evaluates this cost function and updates the parameters iteratively to minimize the objective: $\boldsymbol{\theta}^* = \underset{\boldsymbol{\theta}}{\arg\min}\, C(x,\boldsymbol{\theta})$. This quantum-classical loop is continued until convergence, combining quantum computational advantages with classical optimization techniques (Farhi & Neven, 2018).

### 2.2 QUANTUM MODELS AS FOURIER SERIES

Often, multiple FMs and ansätze are combined into a parameterized quantum circuit $U(x,\boldsymbol{\theta})$, alternating $L$ angle encoding FMs $S(x)$ with $L+1$ trainable ansatz layers $W(\boldsymbol{\theta})$:

$$U(x,\boldsymbol{\theta}) = W^{(L)}(\boldsymbol{\theta_{L+1}})S(x)W^{(L)}(\boldsymbol{\theta_L})\dots S(x)W^{(0)}(\boldsymbol{\theta_1}) \tag{1}$$

As demonstrated by Schuld et al. (2021), a quantum model $f_\theta(x)$ utilizing quantum circuit $U(x,\boldsymbol{\theta})$ and observable $M$ naturally represents a partial Fourier series:

$$f_{\boldsymbol{\theta}}(x) = \langle 0| U^\dagger(x,\boldsymbol{\theta})MU(x,\boldsymbol{\theta})|0\rangle \tag{2}$$

$$= \sum_{\omega\in\Omega} c_\omega e^{i\omega x} \tag{3}$$

The repeated application of FMs $S(x) = e^{ixH}$, where $H$ is a Hamiltonian, generates Fourier frequencies $\omega$ that collectively form the frequency spectrum $\Omega$. Due to the sequential arrangement of FMs on the same qubit across different layers, this configuration is termed a "serial" architecture. An equivalent "parallel" architecture can generate the same frequency spectrum $\Omega$ by distributing the $L$ FMs across $L$ distinct qubits:

$$U_p(x,\boldsymbol{\theta}) = W_p^{(1)}(\boldsymbol{\theta_2})S_p(x)W_p^{(0)}(\boldsymbol{\theta_1}) \tag{4}$$

operating on the initial state $|0\rangle^{\otimes L}$ with Pauli-Z measurement on one or more qubits. We focus on the standard choice of Hamiltonian $H = \frac{1}{2}\sigma$ for angle encoding FMs, where $\sigma \in \{\sigma_x, \sigma_y, \sigma_z\}$

represents a Pauli matrix. This allows $S(x)$ to be implemented as a single-qubit rotational gate $R \in \{R_x, R_y, R_z\}$ with $R = e^{-i\frac{x}{2}\sigma}$. For the parallel architecture, the $L$-qubit FM $S_p(x)$ represents a tensor product of single-qubit FMs. Since Pauli rotation gates commute when acting on different qubits, this can be diagonalized as:

$$S_p(x) = e^{-i\frac{x}{2}\sigma_L} \otimes \ldots \otimes e^{-i\frac{x}{2}\sigma_1} \tag{5}$$

$$= V_L e^{-i\frac{x}{2}\sigma_z} V_L^\dagger \otimes \ldots \otimes V_1 e^{-i\frac{x}{2}\sigma_z} V_1^\dagger \tag{6}$$

$$= V \exp\left(-i\frac{x}{2}\sum_{q=1}^{L}\sigma_z^{(q)}\right) V^\dagger \tag{7}$$

$$= V e^{-ix\Sigma} V^\dagger \tag{8}$$

where $\sigma_z^{(q)}$ denotes the diagonal $L$-qubit operator acting as $\sigma_z$ only on the $q$th qubit, $\Sigma = \text{diag}(\lambda_1, \ldots, \lambda_{2^L})$ and $V$ contains the eigenvectors of $H$ as columns. The frequency spectrum $\Omega$ comprises all pairwise differences between the eigenvalues $\lambda_i$, yielding $(2^L)^2 = 4^L$ total frequencies, of which $2L+1$ are unique (Schuld et al., 2021). Beyond purely serial or parallel configurations, hybrid architectures combining $l_s$ serial and $l_p$ parallel FMs can achieve identical frequency spectra, provided $l_s \cdot l_p = L$ (Holzer & Turkalj, 2024).

## 2.3 Encoding Strategies and Frequency Spectrum Generation

Quantum models with angle encoding serve as universal function approximators under certain conditions, as demonstrated by Schuld et al. (2021). The expressivity of these models is fundamentally linked to their frequency spectrum: larger frequency spectra enable better approximation of arbitrary functions. This relationship is illustrated in Figure 1, where we compare the approximation of a square wave target function using 1 (positive) frequency versus 3 and 9, demonstrating progressive improvement with increased spectral richness.

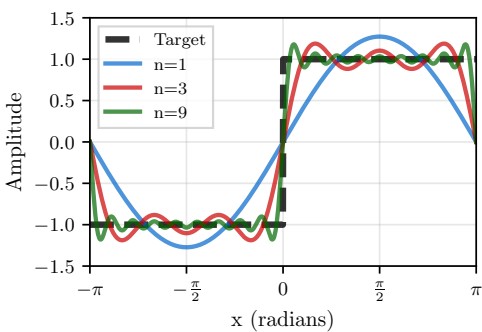

Figure 1: Approximation of the square wave target function by a Fourier series with 1, 3 and 9 (positive) frequencies. The approximation improves in general with the availability of more frequencies.

The frequency spectrum is generated through pairwise differences of eigenvalues in the diagonal matrix $\Sigma = \text{diag}(\lambda_1, \ldots, \lambda_{2^L})$, as established in Equation (8). However, both the eigenvalues and their pairwise differences inherently contain redundancies that limit the effective frequency spectrum. To address this limitation, we can multiply individual FM Hamiltonians $\sigma_i$ by prefactors $p_i$, which correspondingly scales the eigenvalues in $\sigma_i$ by the same prefactor. This scaling strategy enables the (partial) elimination of redundancies arising from degenerate eigenvalues in $\Sigma$ and identical differences between them. Ternary encoding represents a particularly efficient approach. This method employs prefactors of $p_i = 3^i$ with $i \in \{0, L-1\}$, achieving exponential growth in the number of unique frequencies per repeated encoding gate (Peters & Schuld, 2023; Kordzanganeh et al., 2023; Shin et al., 2023). It maximizes the utilization of the available $(2^L)^2$ frequency space by generating a dense frequency spectrum containing $3^L$ unique frequencies—the largest possible cardinality for separable encoding generators while simultaneously minimizing redundant frequencies. This comprehensive frequency spectrum provides significant practical advantages, enabling the coverage of a substantial number of frequencies in the target function using a relatively compact set of encoding gates:

$$|\Omega| = 3^L = \left|\left\{-\left\lfloor\frac{3^L}{2}\right\rfloor, -\left\lfloor\frac{3^L}{2}\right\rfloor + 1, \ldots, 0, \ldots, \left\lfloor\frac{3^L}{2}\right\rfloor - 1, \left\lfloor\frac{3^L}{2}\right\rfloor\right\}\right| \tag{9}$$

In the following, we will focus on unary and ternary encoding, with prefactors of 1 and $3^i$ respectively.

# 3 PARAMETER REQUIREMENTS FOR FREQUENCY CONTROL AND SCALING CHALLENGES

A fundamental constraint in quantum machine learning emerges from the relationship between frequency control and parameter requirements. For a quantum model to fully utilize its frequency spectrum, each unique frequency must possess independent control through at least one dedicated parameter (Schuld et al., 2021):

$$p \geq |\Omega| \quad \text{for } \boldsymbol{\theta} \in \mathbb{R}^p \tag{10}$$

As will be seen later in Section 5, this needs to be understood as necessary requirement since controlling the unique frequencies only creates a lower bound for the parameters needed. This parameter-frequency relationship creates a critical scaling challenge that is often underestimated in QML ansatz design. Target functions containing exponentially many frequency components necessitate quantum circuits with correspondingly exponential numbers of parameters to achieve accurate representation. The scaling challenge is compounded by fundamental constraints on parameter addition. Even with sufficient individual parameters, the number of linearly independent Fourier coefficients remains bounded by the DLA dimension—for example, a single qubit provides only three linearly independent parameters regardless of circuit depth. This limitation is further compounded by the difficulty of creating additional individual parameters in the first place: consecutive (parameter dependent) rotations on the same qubit axis collapse into a single effective parameter (Nielsen & Chuang, 2002). Only when non-linearity through feature maps or entangling gates separates parameter rotations can additional parameters be trained independently to different values.

## 3.1 MULTI-DIMENSIONAL EXTENSIONS AND MIXED FREQUENCIES

The extension to multi-dimensional datasets introduces additional complexity through the emergence of mixed frequencies. For datasets containing $d$ features, repeated data encodings generate individual frequency spectra $\Omega_i$ for each feature $i$. Interactions between features within the circuit allow each frequency component from one feature's spectrum to combine with frequency components from all other features. This coupling mechanism generates mixed frequencies, and the resulting overall frequency spectrum $\Omega$ becomes the Cartesian product of all individual frequency spectra (Holzer & Turkalj, 2024):

$$\Omega = \Omega_1 \times \ldots \times \Omega_d$$
$$|\Omega| = \prod_{i=1}^{d} |\Omega_i| \tag{11}$$

This multi-dimensional formulation reveals a double exponential scaling challenge. While each individual feature's frequency spectrum already grows exponentially with $r$ repeated ternary feature maps ($|\Omega_i| = 3^r$), the entangled multi-dimensional model experiences an additional layer of exponential growth proportional to the number of features. The total frequency spectrum cardinality scales as $(3^r)^d$, enabling comprehensive coverage of extensive frequency spaces that include arbitrary mixed frequency combinations between features. However, this representational richness comes at a steep parameter cost. Following the constraint established in Equation (10), every frequency in the spectrum requires individual coefficient control for independent utilization. In the multi-dimensional case with mixed frequencies, this translates to a parameter requirement of at least $(3^r)^d$—a doubly exponential scaling that presents significant practical challenges for high-dimensional datasets.

## 3.2 AVOIDING SPURIOUS LOCAL MINIMA WITH ANSATZ OVERPARAMETERIZATION

The $L + 1$ ansatz layers $W^{(i)}$ in Equation (1) are typically constructed from one or more identical blocks $W_j^{(i)}$ to facilitate scalability, as detailed in Section D. Each block operates on multiple qubits, but is composed of elementary gates acting on one- or two-qubit subsystems: $W_j^{(i)} = \prod_k W_{j,k}^{(i)} = \prod_k e^{a_k}$, where the set of generators is defined as $\mathcal{G} = \{a_k\}_{k=1}^{K}$ and $K$ represents the total number of generators within an ansatz block (Wiersema et al., 2024). A widely adopted ansatz architecture

is the Hardware Efficient Ansatz (HEA) (Kandala et al., 2017), which combines parameterized rotational gates with entanglement gates such as CNOTs. In this framework, the ansatz generators $\{a_k\}$ consist of single-qubit Pauli rotations $\sigma \in \{\sigma_x, \sigma_y, \sigma_z\}$ or the identity $I$, and tensor products thereof for two-qubit operations, such as $\pi/4(Z \otimes I - Z \otimes X)$ for the CNOT gate. Since combinations of generators can produce additional generators through commutation relations, it becomes essential to characterize the DLA $\mathfrak{g}$, which describes the complete set of unitary evolutions accessible to a given generator set. The DLA is defined through the Lie closure of the generators:

$$\mathfrak{g} = \mathrm{span}\langle ia_1, \ldots, ia_K \rangle_{Lie} \tag{12}$$

where the Lie closure is formed by repeated nested commutators of the generators in $\mathcal{G}$. The DLA fundamentally determines the set of reachable unitaries $U(x, \boldsymbol{\theta})$ that can act on the encoded input states $\Psi_0(x)$, thereby constraining the accessible quantum state space $\Psi(x, \boldsymbol{\theta})$. Work from Larocca et al. (2023) reveals that spurious local minima can be avoided when the number of ansatz parameters exceeds the dimension of the DLA, enabling the quantum model to explore all directions within the state space spanned by the ansatz generators. This establishes an additional parameter requirement:

$$p \geq \dim(\mathfrak{g}) \tag{13}$$

While this condition is sufficient, numerical investigations by Larocca et al. (2023) demonstrate that the required parameter count often needs only to approximate this bound to effectively eliminate spurious local minima. For the commonly employed HEA with universal gate sets, the DLA dimension scales exponentially with the number of qubits (Larocca et al., 2023):

$$\dim(\mathfrak{g}) = 4^n \tag{14}$$

where $n$ denotes the qubit count. This exponential parameter requirement severely limits trainability, making models with more than approximately 5 qubits already computationally challenging in simulations. In contrast, less expressive ansatz architectures offer more favorable scaling properties. For instance, the Hamiltonian variational ansatz studied by Wiersema et al. (2020) exhibits linear scaling with $\dim(\mathfrak{g}) = \frac{3}{2}n$, providing a more tractable alternative for larger quantum systems.

### 3.3 SUMMARY OF REQUIREMENTS

The analysis presented in the preceding sections reveals three fundamental conditions on the number $p$ of ansatz parameters that must be satisfied for effective quantum machine learning model training:

1. **Frequency coverage condition**: The frequencies $\omega_t$ present in the target function must be contained within the model's frequency spectrum $\Omega_m$: $\omega_t \in \Omega_m$. This ensures that the quantum model possesses the necessary spectral components to represent the target function.

2. **Frequency control condition**: For complete utilization of all frequencies in the model spectrum, the number $p_{li}$ of individually trainable ansatz parameters must equal or exceed the cardinality of the frequency spectrum: $p_{li} \geq |\Omega|$. This condition guarantees individual control over each unique frequency component.

3. **Optimization landscape condition**: To avoid spurious local minima during training, the number of individually trainable parameters must equal or exceed the dimension of the DLA: $p_{li} \geq \dim(\mathfrak{g})$. This ensures sufficient flexibility to explore the complete accessible state space.

These three conditions collectively establish the minimum parameter requirements for quantum machine learning models, with the effective parameter count being determined by $p_{li} = \max\{|\Omega|, \dim(\mathfrak{g})\}$, subject to the constraint that the target function frequencies are representable within the chosen encoding scheme.

## 4 FREQUENCY SELECTION AND NEAR-ZERO WEIGHT INITIALIZATION

### 4.1 PARAMETER INSUFFICIENCY

To investigate parameter sufficiency requirements, we employ one-dimensional whitebox target functions that allow precise control over theoretical parameter needs. We examine both single-qubit

serial encoding (prone to linear parameter dependencies and therefore failing to generate additional individual parameters when additional parameters are added between FMs as they can be collapsed into 3 linearly independent parameters) and multi-qubit parallel approaches with and without ancilla qubits (capable of generating additional individually trainable parameters). Detailed circuit diagrams for the various architectures are provided in Section D. By comparing unary and ternary encoding schemes which generate different numbers of non-unique frequency components, we study the parameter requirement scaling caused by non-unique frequencies.

## 4.2 WEIGHT INITIALIZATION

For architectures with redundant non-unique frequencies, we implement near-zero weight initialization through scaling the randomly initialized weights by factors of $0.1$ and $0.01$ to facilitate suppression of unnecessary duplicate frequency coefficients. For multi-qubit implementations, we increase circuit expressivity by using three overlapping 2-qubit Special Unitary gates, each with 15 parameters ((PennyLane, 2025); for the circuit diagram please see Figure 19 in the Appendix) gates to improve loss landscape smoothness under near-zero initialization conditions.

## 4.3 FREQUENCY SELECTION

Quantum models employing ternary encoding generate the largest dense frequency spectrum achievable for separable encoding generators. While this comprehensive spectral coverage ensures that potential target function frequencies are available, it necessitates exponential parameter scaling that often renders practical implementation infeasible. However, when the target function's spectral composition is known a priori, this complete frequency coverage becomes unnecessary and computationally wasteful. By selecting prefactors that deviate from the standard ternary sequence—either through manual selection, systematic trial-and-error approaches, or dedicated classical optimization—we can leverage Equation (8) to construct sparse frequency spectra containing gaps that ideally encompass only the frequencies present in the target function. To illustrate the reduction in spectral complexity by frequency selection, consider a two-feature-map configuration with $R_x$ encoding gates and prefactors of 3 and 9. This yields the eigenvalue matrix: $\Sigma = \frac{1}{2}\text{diag}(3 + 9, 3 - 9, -3 + 9, -3 - 9) = \frac{1}{2}\text{diag}(12, -6, 6, -12)$. Computing the unique pairwise differences produces the sparse frequency spectrum $\Omega = \{-12, -9, -6, -3, 0, 3, 6, 9, 12\}$ containing only 9 frequencies. In contrast, the equivalent dense ternary spectrum with prefactors $\{1, 3, 9\}$ that would naturally contain the target frequencies $\{3, 6, 9, 12\}$ spans the much larger range $\{-13, -12, \ldots, -1, 0, 1, \ldots, 12, 13\}$ with 27 frequencies.

# 5 MITIGATING EXPONENTIAL PARAMETER REQUIREMENTS

## 5.1 EXPERIMENTAL SETUP

Our experimental approach proceeds in two stages. First, we evaluate several circuit architectures using a 1D target function to identify which configurations can satisfy the parameter requirements established in Section 3.3. Subsequently, we extend the investigation to two-dimensional target functions using the architectures that successfully meet these requirements. A detailed description of the experimental setup can be found in Section A in the Appendix.

### 5.1.1 TARGET FUNCTION DESIGN

To systematically demonstrate the parameter insufficiency problem, we employ white box target functions in the form of real-valued partial Fourier series $t \colon [-\pi, \pi]^d \to \mathbb{R}$. The target frequencies are chosen as $\{3, 6, 9, 12\}$ and $\{3, 6, 9, 12\} \times \{3, 6, 9, 12\}$ for the 1D and 2D case, respectively. The coefficients for the individual frequencies randomly sampled from the interval $[0, 1)$ to generate 10 distinct target function instances that differ only in their coefficients while maintaining identical spectral characteristics. This approach enables precise knowledge of the required target frequencies, ensuring that our quantum models possess the necessary spectral components. Under these controlled conditions, near-perfect fitting performance should be achievable, making any systematic failures attributable to parameter limitations rather than spectral inadequacy. For the experimental evaluation, mean squared error serves as the optimization objective, while $R^2$ scores provide the

primary performance evaluation metric. Results are summarized using interquartile ranges (IQRs), with error bars representing the spread between the 25th and 75th percentiles around the mean $R^2$ score. Further details on the experimental setup can be found in Section A.

### 5.1.2 FREQUENCY GENERATORS

For all target functions, we exploit our complete knowledge of the target frequency spectrum to design optimal encoding strategies. This expert knowledge enables two distinct approaches: dense frequency coverage and strategic frequency selection through specialized prefactors which we employ for 2D cases with the circuit architectures identified as suitable with the dense spectra in the 1D experiments.

**1D Target Functions:** We implement dense frequency spectrum generation using two alternative configurations. The first employs 12 unary FMs with prefactors of 1, yielding the spectrum $\Omega = \{-12, -11, \dots, 0, \dots, 11, 12\}$ containing 25 frequencies. The second utilizes only 3 ternary FMs with prefactors $\{1, 3, 9\}$, producing the spectrum $\Omega = \{-13, -12, \dots, 0, \dots, 12, 13\}$ with 27 frequencies with a far shallower circuit.

**2D Target Functions:** We compare selected frequency approaches against dense frequency methodologies to demonstrate the effectiveness of our parameter reduction strategy. The selected frequency models employ prefactors of 3 and 9, generating the sparse spectrum $\Omega = \{-12, -9, -6, -3, 0, 3, 6, 9, 12\}$ containing only 9 frequencies. In contrast, the dense frequency models utilize ternary encoding with prefactors $\{1, 3, 9\}$, producing the comprehensive spectrum $\Omega = \{-13, -12, \dots, 0, \dots, 12, 13\}$ with 27 frequencies.

### 5.2 CIRCUIT ARCHITECTURES AND PARAMETER REQUIREMENT VIOLATIONS FOR 1D TARGET FUNCTIONS

We begin our investigation with 1D partial Fourier series using model configurations that provide dense frequency spectrum coverage of all target frequencies. The results in Figure 2a largely

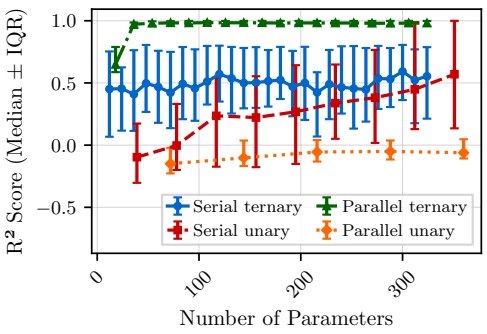 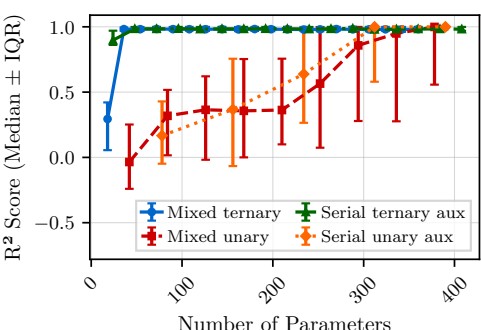

(a) 1D partial Fourier series fitted with standard serial and parallel architectures.

(b) 1D partial Fourier series fitted with enhanced serial architectures.

Figure 2: $R^2$ scores and IQRs for increasing numbers of model parameters (introduced by adding additional ansatz layer between FMs) demonstrate how circuit architectures influence fitting quality for the 1D partial Fourier series target function.

align with our theoretical parameter requirements from Section 3.3 (detailed individual boxplots are provided in Figure 5 in the Appendix). The parallel ternary architecture on 3 qubits quickly achieves sufficient nominal parameters through additional ansatz layers to satisfy both frequency control requirements (27 frequencies) and DLA constraints (64 dimensions). Conversely, the parallel unary architecture cannot generate adequate parameters for its 12-qubit ansatz DLA requirement of $4^{12} \approx 16.8$ million, while the serial ternary architecture lacks sufficient individually trainable parameters to control 27 frequencies, as additional ansatz layers on a single qubit between two feature maps do not create additional individual parameters. However, the serial unary architecture presents an unexpected case and is examined in more detail in Section B in the Appendix. It contains 39 parameters that are generated by 13 ansatz layers with 3 parameters each. These 39 parameters cannot

be collapsed into just 3 parameters as each ansatz layer is separated from the next ansatz layer by a data dependent (non-linear) rotational gate $R_x(x)$ and are therefore able to be trained to individual values. With 39 parameters, it satisfies both frequency (25) and DLA (3) requirements already with only a single ansatz layer between the individual unary FMs, yet fails to achieve perfect fitting. This discrepancy reiterates that the requirements stated in Section 3.3 only serve as lower boundaries and motivates our second experimental investigation.

The results in Figure 2b reveal that both the number of serial layers and parameter interactions play crucial roles (detailed boxplots in Figure 6 in the Appendix). When unary models employ ansätze spanning 2 qubits, perfect fits become achievable once parameter counts exceed approximately 380. This holds whether adding a second qubit with FMs (models described as "mixed") or with only rotational ansatz gates and the ability to introduce CNOT gates (models described as "aux"). Remarkably, ternary models achieve perfect fits with as few as 36 parameters when a second qubit is added. This substantial gap between unary and ternary parameter requirements illuminates a more subtle parameter constraint: control over non-unique (degenerate) frequencies is also essential. Ternary and unary create almost the same number of unique frequencies with 27 and 25 respectively. The big difference between the two approaches lies in the number of non-unique frequencies which is $64 - 27 = 37$ for the ternary and $4^{12} - 25 \approx 17M$ for the unary approach. The total number of frequencies—not merely the unique ones— therefore influences overall parameter demands. While many degenerate frequencies can share parameter control (when they are set to 0 for example), insufficient parameters to manage unique frequencies alone may not guarantee success. Exponential encoding offers the advantage of minimizing non-unique frequencies, reducing the maximum additional parameters needed for degenerate frequency control and enabling coverage of all model frequencies at $4^l$ (where $l$ is the number of FMs) with fewer total parameters.

## 5.3 NEAR-ZERO WEIGHT INITIALIZATION

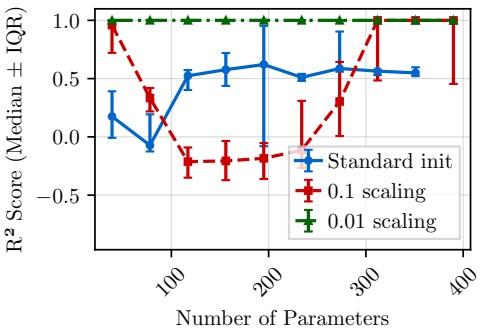 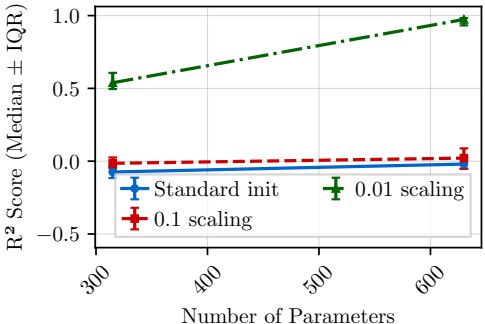

(a) IQR comparison of $R^2$ scores for the serial unary encoding architecture on 1 qubit for the 1D target functions with random weight parameter initializations in $[0, 2\pi)$ (Standard init), $[0, 0.2\pi)$ (init 0.1 scaling) and $[0, 0.02\pi)$ (init 0.01 scaling).

(b) IQR comparison of $R^2$ scores for the serial unary encoding architecture on 4 qubits for the 2D target functions with random weight parameter initializations in $[0, 2\pi)$,(Standard init), $[0, 0.2\pi)$ (init 0.1 scaling) and $[0, 0.02\pi)$ (init 0.01 scaling).

Figure 3: $R^2$ scores and IQRs for increasing numbers of model parameters (introduced by adding additional ansatz layer between FMs) demonstrate how the optimization process can be accelerated by a weights initialization near 0.

Since non-unique frequencies are redundant for our target functions, their corresponding coefficients should ideally be driven to zero during training, while coefficients for the target frequencies must match (or sum to) their respective target values. This insight motivates initializing parameters close to zero for which some theoretical background is provided in Section B in the Appendix. The results in Figure 3 demonstrate clear advantages for this initialization strategy when random weight initialization is scaled by a factor of 0.01. More detailed boxplots are provided in Figure 8 and Figure 7 in the Appendix. For the 1D case, $R^2$ scores approaching 1 can be achieved with a single ansatz layer between feature maps. The 2D case, however, required two modifications to the standard setup before reaching near optimal $R^2$ scores with 2 ansatz layers: (i) The circuit's expressivity was increased by replacing the combination of rotational gates and CNOT gates with three overlapping

2-qubit Special Unitary gates, each with 15 parameters ((PennyLane, 2025); for the circuit diagram please see Figure 19 in the Appendix). (ii) The number of samples per dimension was increased from 25 to 50, yielding a total of 2,500 samples, of which 2,000 were used for training.

## 5.4 FREQUENCY SELECTION TECHNIQUES

The experimental results in Figure 4 demonstrate the substantial advantages of strategic frequency selection for two-dimensional target functions. Dense frequency models require significantly more parameters to achieve the near-perfect fitting performance that their comprehensive spectral coverage enables. In contrast, the selected frequency model achieves superior $R^2$ scores and reaches near-optimal performance with considerably fewer parameters than their dense counterparts. This performance advantage stems from the targeted approach of focusing exclusively on frequencies present in the target function. By restricting the model spectrum to essential frequencies only, the number of feature maps can be reduced from 3 to 2. This reduction yields a dual benefit: both unique and non-unique (degenerate) frequency counts decrease to 9 and 16 respectively, enabling comprehensive control over all model

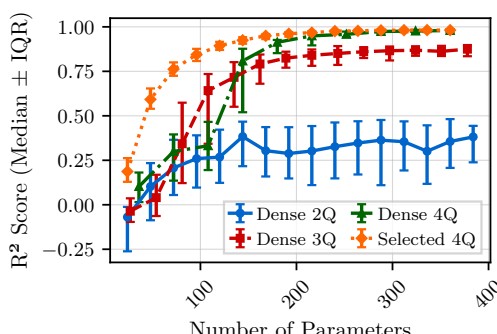

Figure 4: Comparison of $R^2$ scores between dense encoding architectures on architectures with various qubits and the selected frequency approach for the 2D partial Fourier series target function.

frequencies with significantly fewer parameters compared to 27 and 64 in the dense frequency case. The strategic elimination of irrelevant spectral components thus directly translates to improved parameter efficiency and enhanced model trainability, validating our frequency selection methodology for multi-dimensional quantum machine learning applications. The significant difference becomes apparent already in the modest reduction by 1 FM. For more substantial differences in the number of FMs and consequently the number of unique / total frequencies, the trainability difference will become even more pronounced.

## 6 DISCUSSION

**Interpretation and Related Work:**  Our white box experiments systematically demonstrate the fundamental limitations of current quantum models with angle encoding when deployed with extensive frequency spectra. The required parameter counts rapidly exceed current hardware capabilities, creating a practical bottleneck for quantum machine learning applications. Near-zero weight initialization demonstrates a promising technique for accelerating optimization when many redundant model frequencies are absent from the target function. Our selected frequency approach circumvents these limitations through strategic reduction of frequencies rather than exhaustive dense frequency space exploration. As an alternative approach to address parameter scalability in quantum machine learning Jaderberg et al. (2024) suggested allowing the frequency prefactors to be trained as learnable parameters, providing another pathway to optimize spectral properties. Wiedmann et al. (2024) pointed out that in some ansätze, the parameters can be systematically constrained to 0, effectively removing the corresponding frequency from the model spectrum—a double-edged mechanism that can eliminate unnecessary frequencies but may also remove essential ones.

**Limitations and Scope of Applicability:**  Several important limitations constrain the applicability of our approach. First, our experimental targets were partial Fourier series themselves, ensuring perfect representability within our chosen model class. Real-world datasets may exhibit structures less amenable to Fourier series approximation, limiting achievable accuracy to what the available parameter budget and corresponding frequency set permit. Second, our methodology assumes complete knowledge of the target function's frequency composition—a simplification suitable for controlled experiments but unrealistic for practical applications. In real-world settings, relevant frequencies must be identified through domain expertise or data-driven analysis. One approach applies classi-

cal Fourier analysis to extract dominant frequencies for each feature dimension (Wiedmann et al., 2024), though this preprocessing introduces computational overhead and potential approximation errors. To extend our approach beyond controlled settings, we propose performing 1D Fourier analysis on slices of the multi-dimensional dataset to identify the most relevant frequencies per dimension. Combinatorial optimization can then determine the minimal set of feature maps required to generate these frequencies. The exponential scaling of ternary encoding makes this optimization tractable: just seven ternary layers already generate more than 1,000 distinct positive frequencies, providing broad coverage with few feature maps. Finally, even with carefully selected frequency sets per dimension, the explosive growth of mixed frequencies rapidly leads to infeasible parameter requirements. For data sets exhibiting negligible interdependencies between feature subsets, a dimensional separation strategy may enhance tractability. This approach involves modeling and measuring interconnected subsets independently before classical combination for parameter optimization. However, further research is required to extend this technique beyond the limited cases where such independent feature blocks naturally occur.

## 7 CONCLUSION

Quantum machine learning faces exponential parameter scaling when representing target functions with dense Fourier spectra across multiple dimensions. This challenge creates parameter requirements that rapidly exceed current hardware capabilities, limiting QML's practical applicability to real-world problems. Through controlled whitebox experiments, this work identifies these fundamental limitations and addresses them via frequency selection and near-zero weight initialization. By leveraging a priori knowledge of essential frequencies in each dimension, we demonstrate substantial parameter reduction while mitigating exponential scaling. Near-zero weight initialization proves highly effective for one- and two-dimensional cases, especially for unary models with a large amount of redundant model frequencies. Our contributions provide insights into QML failure modes and practical techniques for implementing quantum machine learning solutions that respect hardware constraints. As quantum hardware continues to evolve, these parameter-efficient approaches will be essential for bridging the gap between theoretical QML advantages and practical quantum computing applications.

## ETHICS STATEMENT

Large language models have been used for the production of this paper to polish writing, debug the code for conducting the experiments and generate the code for plotting the results.

## REPRODUCIBILITY STATEMENT

The circuit diagrams for the various experiments can be found in Section D. The target functions and a high level description of the experimental set up is provided in Section 5.1 and in more detail in Section A. In the supplementary materials we have included a zip file with the Jupyter notebook for one of the experiments each for the 1D and 2D target function case, together with the datasets and the code for generating the plots with the results.

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

## A    APPENDIX A: EXPERIMENTAL SETUP DETAILS

### A.1    TARGET FUNCTION DESIGN

For the one-dimensional case, we define the target function as:

$$t_1(x) = c_0 + \sum_{i=1}^{4} [a_i \cos(\omega_i x) + b_i \sin(\omega_i x)] \tag{15}$$

where $\omega_i \in \{3, 6, 9, 12\}$ represents the target frequency set. The coefficients $c_0, a_i, b_i$ are randomly sampled from the interval $[0, 1)$ to generate 10 distinct target function instances that differ only in their coefficients while maintaining identical spectral characteristics.

The two-dimensional target function extends this framework to include mixed frequency terms:

$$t_2(x, y) = c_0 + \sum_{i=1}^{4} \sum_{j=1}^{4} \Bigg[ a_{ij} \cos(\omega_i^x x) \cos(\omega_j^y y) + b_{ij} \cos(\omega_i^x x) \sin(\omega_j^y y)$$
$$+ c_{ij} \sin(\omega_i^x x) \cos(\omega_j^y y) + d_{ij} \sin(\omega_i^x x) \sin(\omega_j^y y) \Bigg] \tag{16}$$

where $\omega_i^x, \omega_j^y \in \{3, 6, 9, 12\}$ define the frequency components for each dimension. As in the 1D case, all coefficients $c_0, a_{ij}, b_{ij}, c_{ij}, d_{ij}$ are randomly sampled from $[0, 1)$ to create 10 target function realizations with consistent spectral properties but varying amplitudes.

### A.2    DATA PREPARATION AND SCALING

We use evenly spaced points across $[-\pi, \pi]$ for each input dimension. For 1D target functions, we use 50 sample points, while 2D target functions employ a $25 \times 25$ Cartesian grid, yielding 625 total samples, except for the near-zero weight initialization experiments for which we continued to use the 50 sample points per dimension. To ensure compatibility with quantum rotational gate requirements, we apply Scikit-Learn's MinMaxScaler (Pedregosa et al., 2011) to map input values to the interval $[-\pi, \pi]$ and scale target outputs to $[-1, 1]$.

### A.3    IMPLEMENTATION FRAMEWORK AND CIRCUIT ARCHITECTURE

Our experimental implementation utilizes PennyLane (version 0.42.0) (Bergholm et al., 2022) with JAX/JIT compilation (version 0.5.0) (Frostig et al., 2018) for enhanced computational performance. All training and inference operations were executed on a MacBook Pro equipped with an Apple M2 processor, 16 GB memory, and running macOS Sequoia 15.5.

The quantum circuit architecture employs PennyLane's general rotation gates ($qml.rot(\boldsymbol{\theta})$), which decompose into the sequence $R_z(\theta_1) R_y(\theta_2) R_z(\theta_3)$ to provide arbitrary single-qubit operations with three degrees of freedom. These rotation gates are combined with CNOT gates to create the ansatz structure. To ensure parameter effectiveness when measuring exclusively on the bottom qubit (in the computational basis), we implement a specific gate ordering protocol: rotational gates are applied only after receiving the target element of a CNOT gate from the neighboring qubit above, while the outgoing control element of the CNOT gate to the neighboring qubit below is inserted after the rotational gate. The detailed circuit architectures are presented in Section D. For the 2D near-zero weight initialization experiments we employed ansätze with three overlapping 2-qubit Special Unitary gates, each with 15 parameters ((PennyLane, 2025); for the circuit diagram please see Figure 19 in the Appendix)

### A.4    TRAINING CONFIGURATION AND EVALUATION METRICS

We employ DeepMind's Optax Adam optimizer (DeepMind, 2020) with a fixed learning rate of 0.001 for 5000 training iterations. Model validation follows an 80/20 train-test split protocol. To demonstrate the effectiveness of our frequency selection techniques, we deliberately use these standard hyperparameters without further optimization, showing that our approaches achieve expected target function fitting performance where conventional models fail under identical conditions.

All ansatz parameters are initialized using uniform sampling from the interval $[0, 2\pi)$. Mean squared error serves as the optimization objective, while $R^2$ scores provide the primary performance evaluation metric. Results are summarized using interquartile ranges (IQRs), with error bars representing the spread between the 25th and 75th percentiles around the mean $R^2$ score.

Our experimental protocol implements a comprehensive evaluation framework: using a base random seed of 42, we execute 10 independent training runs for each of the 10 target function realizations (which differ only in coefficient values, maintaining identical frequency spectra). This yields 100 total experiments per model configuration, providing robust statistical assessment of model performance across coefficient variations.

## B  APPENDIX B: RELATIONSHIP OF MODEL COEFFICIENTS AND TARGET COEFFICIENTS FOR 1D SEQUENTIAL UNARY MODEL

A central experimental insight emerged from the serial unary model's inability to perfectly fit the 1D target functions in Section 5.2. Notably, this failure occurred despite the model satisfying both frequency and DLA parameter requirements. To provide theoretical context for these difficulties, the following sections examine the relationships between model and target frequencies and coefficients.

### B.1  MODEL FREQUENCIES VERSUS TARGET FREQUENCIES

Our serial quantum model with 12 unary prefactors and a single ansatz layer between the feature maps follows the Fourier representation of Schuld et al. (2021):

$$f(x) = \sum_{\boldsymbol{k},\boldsymbol{j}\in\{1,2\}^{12}} e^{i(\Lambda_{\boldsymbol{k}} - \Lambda_{\boldsymbol{j}})x} a_{\boldsymbol{k},\boldsymbol{j}}, \tag{17}$$

where the multi-indices are defined as $\boldsymbol{k} = (k_1, \ldots, k_{12})$ and $\boldsymbol{j} = (j_1, \ldots, j_{12})$.

This generates the following frequency spectrum $\Omega$:

$$\Omega = \{-12, -11, \ldots, -1, 0, 1, \ldots, 11, 12\}, \tag{18}$$

with multiplicities:

$$H = \{1, 24, \ldots, 2\,496\,144, 2\,704\,156, 2\,496\,144, \ldots, 24, 1\}, \tag{19}$$

for a total of $4^{12}$ non-unique frequencies.

In contrast, the target function contains only a subset of these 25 unique frequencies, each with multiplicity one:

$$\begin{aligned}
t_1(x) &= c_0 + \alpha_1 \cos(3x) + \beta_1 \sin(3x) \\
&\quad + \alpha_2 \cos(6x) + \beta_2 \sin(6x) \\
&\quad + \alpha_3 \cos(9x) + \beta_3 \sin(9x) \\
&\quad + \alpha_4 \cos(12x) + \beta_4 \sin(12x) \\
&= c_0 + c_3 e^{3ix} + c_{-3} e^{-3ix} \\
&\quad + c_6 e^{6ix} + c_{-6} e^{-6ix} \\
&\quad + c_9 e^{9ix} + c_{-9} e^{-9ix} \\
&\quad + c_{12} e^{12ix} + c_{-12} e^{-12ix},
\end{aligned} \tag{20}$$

where $c_{-\omega} = c_\omega^*$ to ensure $t_1(x)$ is real-valued.

The optimization challenge is therefore twofold: (i) suppress all frequencies absent from the target function, and (ii) tune the coefficients of the multiple model frequencies corresponding to each target frequency such that they aggregate to match the single coefficient in the target.

## B.2 MODEL COEFFICIENTS

The coefficient $a_{\boldsymbol{k},\boldsymbol{j}}$ in Equation (17) for each non-unique model frequency is given by Schuld et al. (2021):

$$
a_{\boldsymbol{k},\boldsymbol{j}} = \sum_{i,i'} (W^*(\boldsymbol{\theta}^{(1)}))_{1k_1}^{(1)} (W^*(\boldsymbol{\theta}^{(2)}))_{k_1 k_2}^{(2)} \cdots (W^*(\boldsymbol{\theta}^{(13)}))_{k_{12} i}^{(13)} M_{i,i'}
$$
$$
\times (W(\boldsymbol{\theta}^{(13)}))_{i' j_{12}}^{(13)} \cdots (W(\boldsymbol{\theta}^{(2)}))_{j_2 j_1}^{(2)} (W(\boldsymbol{\theta}^{(1)}))_{j_1 1}^{(1)}, \tag{21}
$$

where $\boldsymbol{\theta}^{(i)} \in \mathbb{R}^3$ for $i = 1, \ldots, 13$.

Using the Pauli-Z observable $M = \mathrm{diag}(1, -1)$, the off-diagonal elements vanish, simplifying to:

$$
\begin{aligned}
a_{\boldsymbol{k},\boldsymbol{j}} =& (W^*(\boldsymbol{\theta}^{(1)}))_{1k_1}^{(1)} (W^*(\boldsymbol{\theta}^{(2)}))_{k_1 k_2}^{(2)} \cdots (W^*(\boldsymbol{\theta}^{(13)}))_{k_{12} 1}^{(13)} \\
&\times (W(\boldsymbol{\theta}^{(13)}))_{1 j_{12}}^{(13)} \cdots (W(\boldsymbol{\theta}^{(2)}))_{j_2 j_1}^{(2)} (W(\boldsymbol{\theta}^{(1)}))_{j_1 1}^{(1)} \\
&- (W^*(\boldsymbol{\theta}^{(1)}))_{1k_1}^{(1)} (W^*(\boldsymbol{\theta}^{(2)}))_{k_1 k_2}^{(2)} \cdots (W^*(\boldsymbol{\theta}^{(13)}))_{k_{12} 2}^{(13)} \\
&\times (W(\boldsymbol{\theta}^{(13)}))_{2 j_{12}}^{(13)} \cdots (W(\boldsymbol{\theta}^{(2)}))_{j_2 j_1}^{(2)} (W(\boldsymbol{\theta}^{(1)}))_{j_1 1}^{(1)}.
\end{aligned} \tag{22}
$$

In our implementation, $W$ is a general rotation gate composed of $R_z(\theta_1) R_y(\theta_2) R_z(\theta_3)$:

$$
\begin{aligned}
W^{(i)} =& \begin{pmatrix} \cos(\theta_1^{(i)}/2) - i \sin(\theta_1^{(i)}/2) & 0 \\ 0 & \cos(\theta_1^{(i)}/2) + i \sin(\theta_1^{(i)}/2) \end{pmatrix} \\
&\times \begin{pmatrix} \cos(\theta_2^{(i)}/2) & -\sin(\theta_2^{(i)}/2) \\ \sin(\theta_2^{(i)}/2) & \cos(\theta_2^{(i)}/2) \end{pmatrix} \\
&\times \begin{pmatrix} \cos(\theta_3^{(i)}/2) - i \sin(\theta_3^{(i)}/2) & 0 \\ 0 & \cos(\theta_3^{(i)}/2) + i \sin(\theta_3^{(i)}/2) \end{pmatrix} \\
=& \begin{pmatrix} W_{11}^{(i)} & W_{12}^{(i)} \\ W_{21}^{(i)} & W_{22}^{(i)} \end{pmatrix}.
\end{aligned} \tag{23}
$$

Each matrix element depends on all three parameters of the respective ansatz layer:

$$
W_{11}^{(i)} = \cos(\theta_2^{(i)}/2) \left[ \cos\left(\frac{\theta_1^{(i)} + \theta_3^{(i)}}{2}\right) - i \sin\left(\frac{\theta_1^{(i)} + \theta_3^{(i)}}{2}\right) \right], \tag{24}
$$

$$
W_{12}^{(i)} = -\sin(\theta_2^{(i)}/2) \left[ \cos\left(\frac{\theta_1^{(i)} - \theta_3^{(i)}}{2}\right) - i \sin\left(\frac{\theta_1^{(i)} - \theta_3^{(i)}}{2}\right) \right], \tag{25}
$$

$$
W_{21}^{(i)} = \sin(\theta_2^{(i)}/2) \left[ \cos\left(\frac{\theta_1^{(i)} - \theta_3^{(i)}}{2}\right) + i \sin\left(\frac{\theta_1^{(i)} - \theta_3^{(i)}}{2}\right) \right], \tag{26}
$$

$$
W_{22}^{(i)} = \cos(\theta_2^{(i)}/2) \left[ \cos\left(\frac{\theta_1^{(i)} + \theta_3^{(i)}}{2}\right) + i \sin\left(\frac{\theta_1^{(i)} + \theta_3^{(i)}}{2}\right) \right]. \tag{27}
$$

To construct the final Fourier series $f(x) = \sum_{\omega \in \Omega} c_\omega e^{i\omega x}$, we sum all coefficients $a_{\boldsymbol{k},\boldsymbol{j}}$ that contribute to the same unique frequency:

$$
c_\omega = \sum_{\substack{\boldsymbol{k},\boldsymbol{j} \in \{1,2\}^{12} \\ \Lambda_{\boldsymbol{k}} - \Lambda_{\boldsymbol{j}} = \omega}} a_{\boldsymbol{k},\boldsymbol{j}}. \tag{28}
$$

Given the multiplicities in Equation (19) and that each coefficient $a_{\boldsymbol{k},\boldsymbol{j}}$ involves all 39 parameters (13 ansatz layers with 3 parameters each), finding suitable parameter combinations becomes challenging.

Two strategies can mitigate this complexity. First, reducing the total number of frequencies while increasing individual parameters—as in the parallel ternary model—improves the chances of finding suitable parameter values. Second, initializing all 39 parameters near zero effectively sets all coefficients $a_{\boldsymbol{k},\boldsymbol{j}} \approx 0$, allowing the optimizer to identify only those parameters that must deviate from zero to generate the target frequencies with correct coefficients. This sparse initialization strategy can significantly accelerate convergence.

## C  APPENDIX C: DETAILED QUANTUM MODEL RESULTS

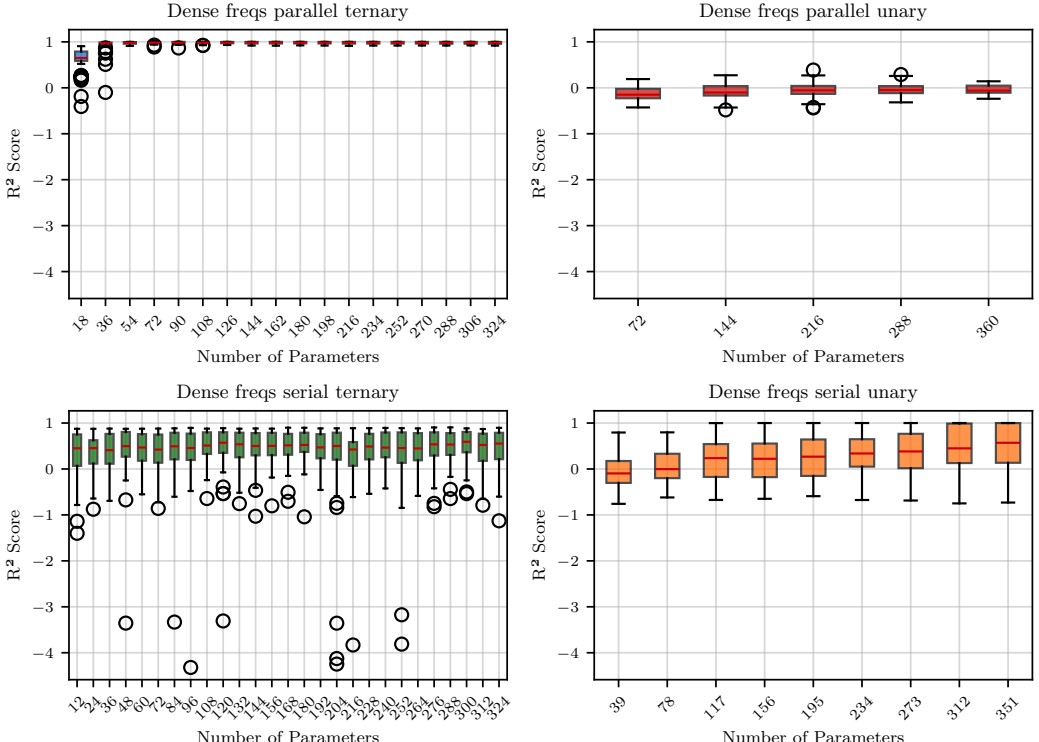

Figure 5: Comparison of $R^2$ scores across serial and parallel encoding architectures for the 1D partial Fourier Series target function.

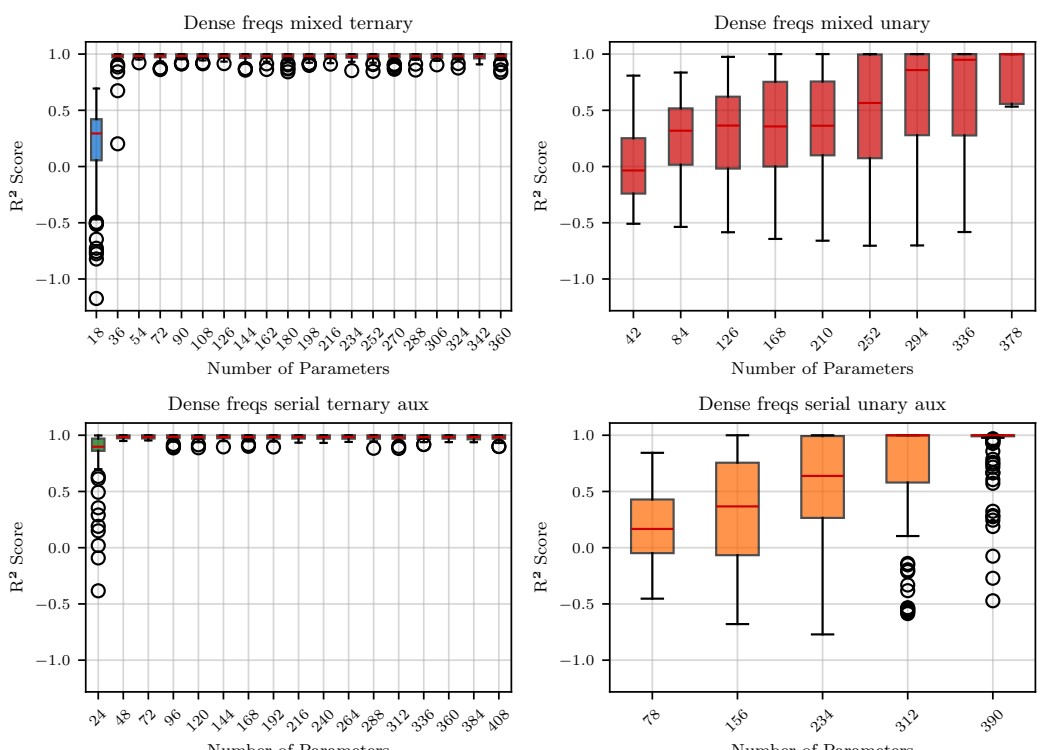

Figure 6: Comparison of $R^2$ scores for the serial encoding architecture with modifications for the 1D partial Fourier Series target function.

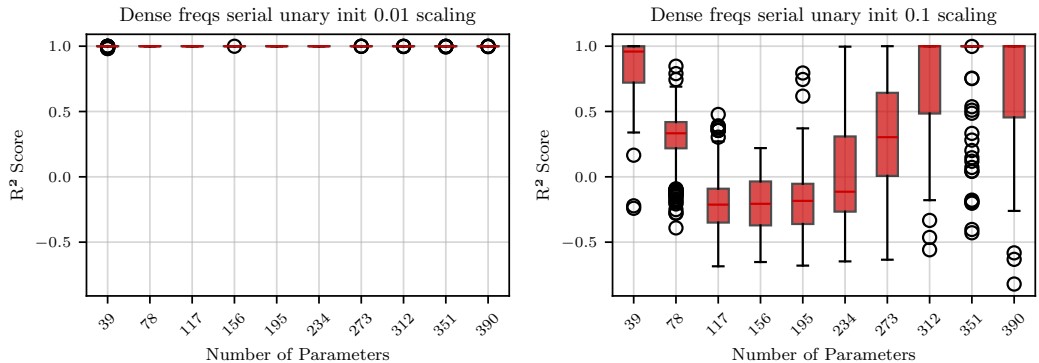

Figure 7: Boxplot comparison of $R^2$ scores for the serial unary encoding architecture on 1 qubit for the 1D target functions with random weight parameter initializations in $[0, 0.2\pi)$ (init 0.1 scaling) and $[0, 0.02\pi)$ (init 0.01 scaling) for the 1D partial Fourier Series target function. The boxplot for the standard random weight parameter initializations in $[0, 2\pi)$ can be found in Figure 5.

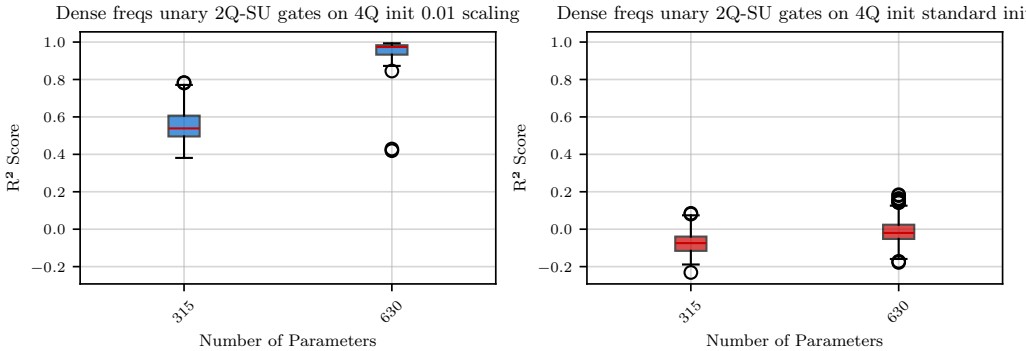

Figure 8: Boxplot comparison of $R^2$ scores for the serial unary encoding architecture with 3 overlapping 2-qubit Special Unitary gates on 4 qubits for the 2D target functions with random weight parameter initializations in $[0, 0.02\pi)$ (0.01 scaling) and $[0, 2\pi)$ (standard init).

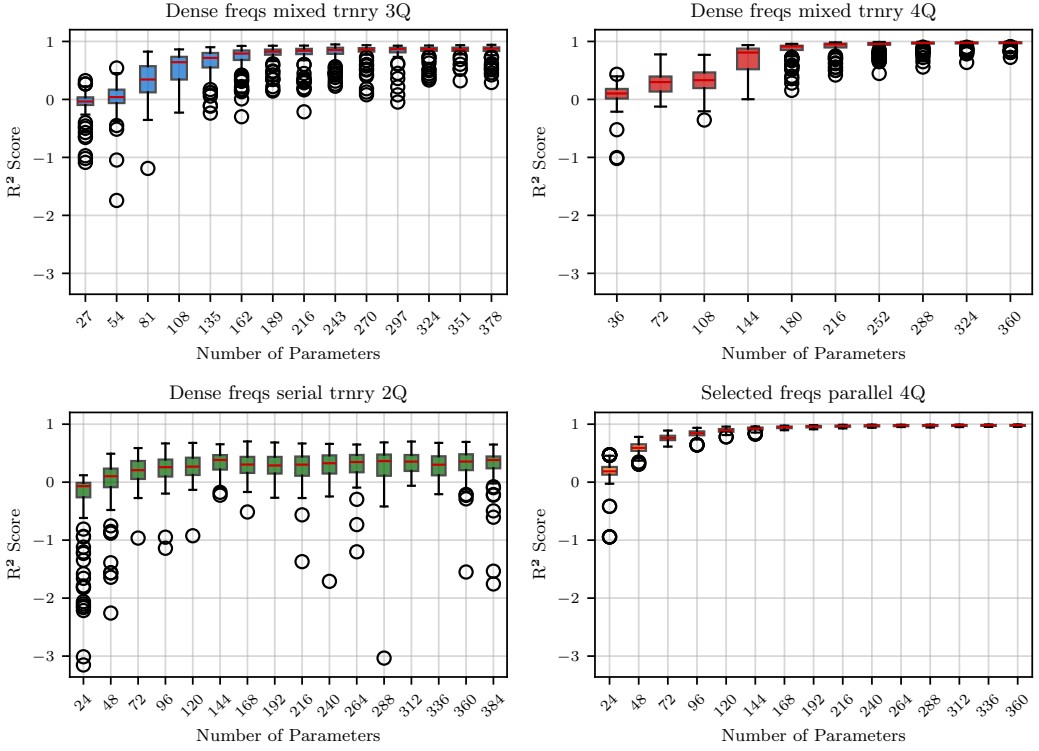

Figure 9: Comparison of $R^2$ scores between dense and selected frequency encoding schemes across serial, mixed and parallel encoding architectures for the 2D partial Fourier Series target function.

# D APPENDIX D: CIRCUIT ARCHITECTURES

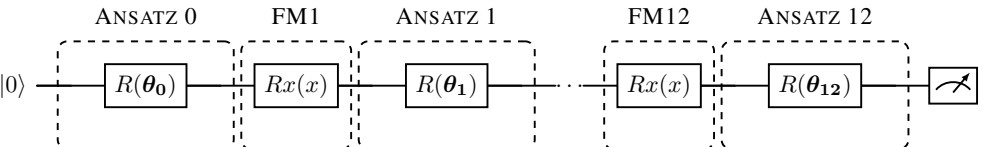

Figure 10: Serial unary circuit architecture for 1D target. To increase the number of parameters, additional general rotational gates $R(\boldsymbol{\theta})$ are added by including additional ansatz layer blocks between the FMs.

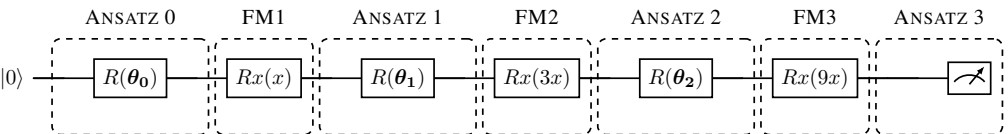

Figure 11: Serial ternary circuit architecture for 1D target. FMs contain prefactors of $3^{l-1}$ for each layer $l = 1, \ldots, 3$. To increase the number of parameters, additional general rotational gates $R(\boldsymbol{\theta})$ are added by including additional ansatz layer blocks between the FMs.

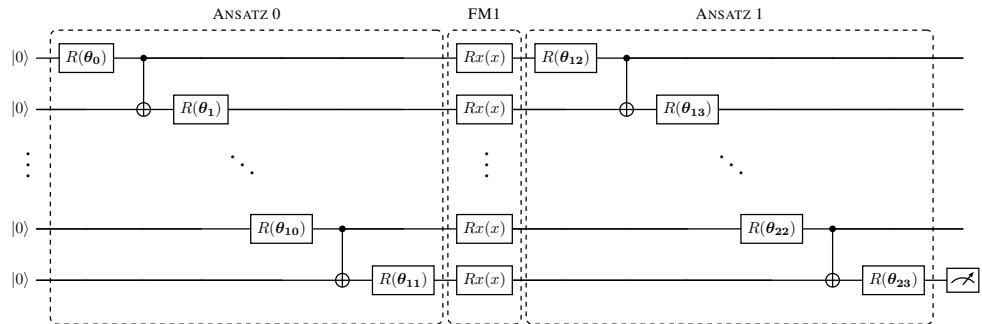

Figure 12: Parallel unary circuit architecture for 1D target. Within each ansatz layer, there is a CNOT connection to the following qubit after the rotational gate. To increase the number of parameters, additional general rotational gates $R(\boldsymbol{\theta})$ are added by including additional ansatz layer blocks between the FMs.

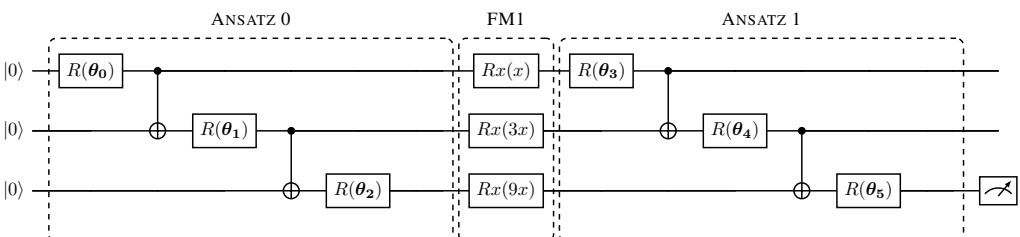

Figure 13: Parallel ternary circuit architecture for 1D target. Within each ansatz layer, there is a CNOT connection to the following qubit after the rotational gate. FMs contain prefactors of $3^{l-1}$ for each vertical layer $l = 1, \ldots, 3$. To increase the number of parameters, additional general rotational gates $R(\boldsymbol{\theta})$ are added by including additional ansatz layer blocks between the FMs.

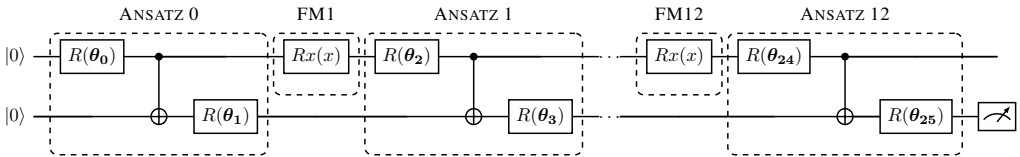

Figure 14: Serial unary aux circuit architecture for 1D target with an extra qubit that allows adding general rotational gates and CNOTs. No FMs are added on the second qubit. To increase the number of parameters, additional general rotational gates $R(\boldsymbol{\theta})$ are added by including additional ansatz layer blocks between the FMs.

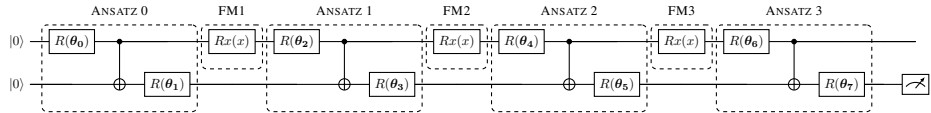

Figure 15: Serial ternary aux circuit architecture for 1D target with an extra qubit that allows adding general rotational gates and CNOTs. No FMs are added on the second qubit. FMs on the first qubit contain prefactors of $3^{l-1}$ for each layer $l = 1, \ldots, 3$. To increase the number of parameters, additional general rotational gates $R(\boldsymbol{\theta})$ are added by including additional ansatz layer blocks between the FMs.

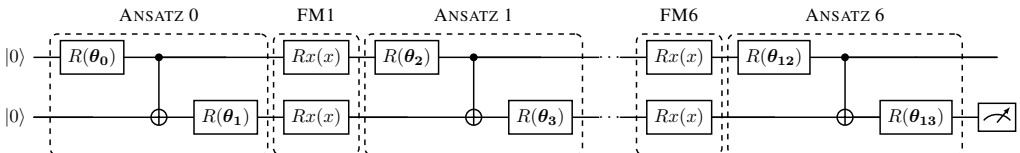

Figure 16: Mixed unary circuit architecture for 1D target with an extra qubit that allows adding general rotational gates, CNOTs and FMs. To increase the number of parameters, additional general rotational gates $R(\boldsymbol{\theta})$ are added by including additional ansatz layer blocks between the FMs.

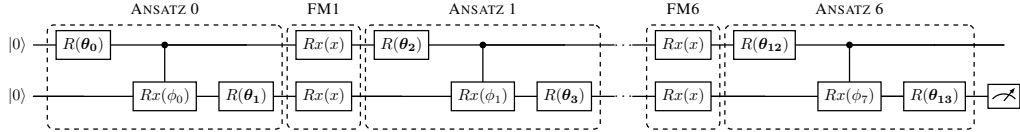

Figure 17: Modified mixed unary circuit architecture for 1D target with an extra qubit that allows adding general rotational gates, entanglement gates and FMs. Instead of CNOT gates, entanglement is created here through parameterized CRX gates. To increase the number of parameters, additional general rotational gates $R(\boldsymbol{\theta})$ are added by including additional ansatz layer blocks between the FMs.

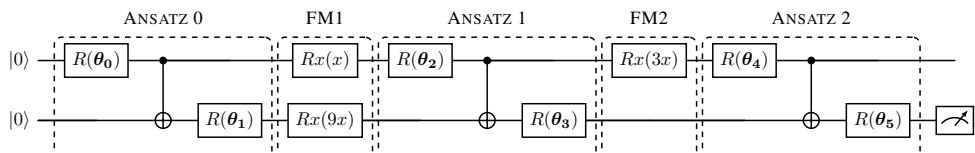

Figure 18: Mixed ternary circuit architecture for 1D target with an extra qubit that allows adding general rotational gates, CNOTs and FMs. FMs contain prefactors of $3^{l-1}$ for each repetition $l = 1, \ldots, 3$. To increase the number of parameters, additional general rotational gates $R(\boldsymbol{\theta})$ are added by including additional ansatz layer blocks between the FMs.

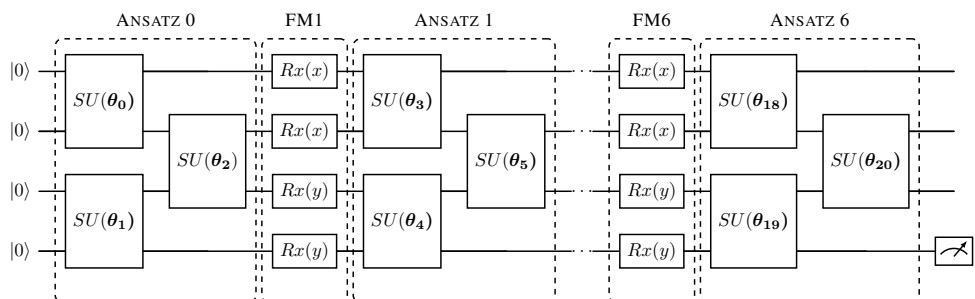

Figure 19: 4 qubit unary circuit architecture with 2-qubit Special Unitary (SU) gates for the near-zero initialization for the 2D target. Within each ansatz layer, there are 3 2-qubit Special Unitary gates. FMs contain prefactors of 1. To increase the number of parameters, additional blocks of 3 Special Unitary gates spanning 2 qubits each are added by including additional ansatz layer blocks between the FMs.

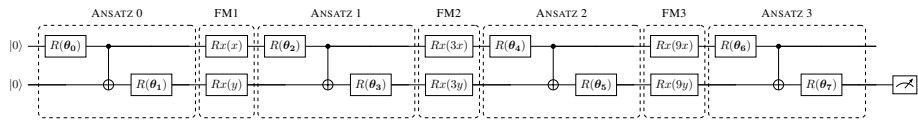

Figure 20: Dense frequencies serial ternary circuit architecture for 2D target on 2 qubits. Within each ansatz layer, there is a CNOT connection to the following qubit after the rotational gate. FMs contain prefactors of $3^{l-1}$ for each consecutive FM in each dimension $l = 1, \ldots, 3$. To increase the number of parameters, additional general rotational gates $R(\boldsymbol{\theta})$ are added by including additional ansatz layer blocks between the FMs.

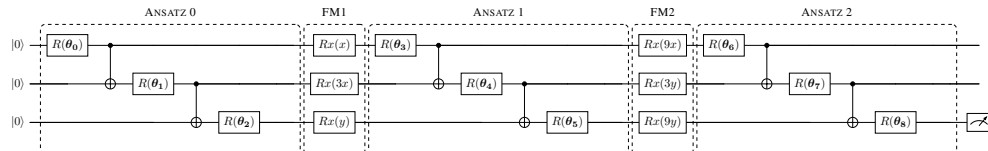

Figure 21: Dense frequencies mixed ternary circuit architecture for 2D target on 3 qubits. Within each ansatz layer, there is a CNOT connection to the following qubit after the rotational gate. FMs contain prefactors of $3^{l-1}$ for each consecutive FM in each dimension $l = 1, \ldots, 3$. To increase the number of parameters, additional general rotational gates $R(\boldsymbol{\theta})$ are added by including additional ansatz layer blocks between the FMs.

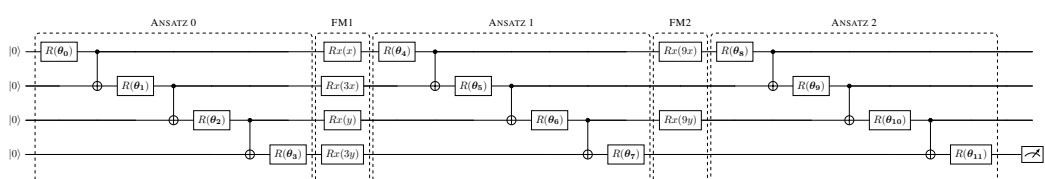

Figure 22: Dense frequencies mixed ternary circuit architecture for 2D target on 4 qubits. Within each ansatz layer, there is a CNOT connection to the following qubit after the rotational gate. FMs contain prefactors of $3^{l-1}$, $l = 1, \ldots, 3$ for each consecutive FM in each dimension. To increase the number of parameters, additional general rotational gates $R(\boldsymbol{\theta})$ are added by including additional ansatz layer blocks between the FMs.

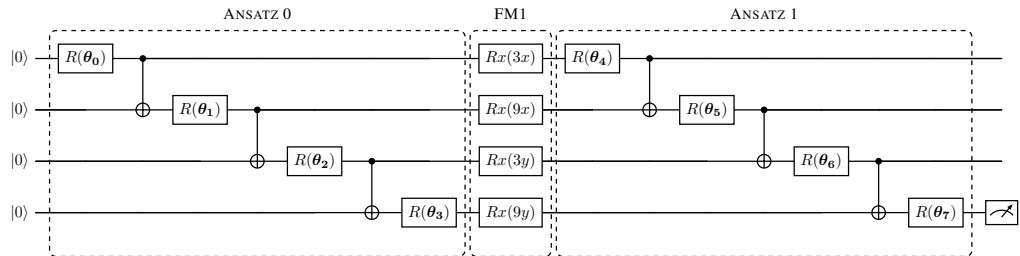

Figure 23: Selected frequencies parallel circuit architecture for 2D target on 4 qubits. Within each ansatz layer, there is a CNOT connection to the following qubit after the rotational gate. FMs contain prefactors of 3 and 9 for each consecutive FM in each dimension. To increase the number of parameters, additional general rotational gates $R(\boldsymbol{\theta})$ are added by including additional ansatz layer blocks between the FMs.

