# OpenReview forum: "Mitigating Exponential Mixed Frequency Growth through Frequency Selection"
_ICLR.cc/2026/Conference — Submitted to ICLR 2026_

### Official Review · Reviewer_f1sr · 2025-11-01

**Soundness:** 3
**Presentation:** 3
**Contribution:** 3
**Rating:** 6
**Confidence:** 2

**Summary:**

This paper considers quantum machine learning and focuses on the angle encoding problem. Due to practical implementation challenges, such as insufficient trainable parameters relative to the model’s frequency content and the ansatz’s dynamic Lie algebra dimension, the paper proposes near-zero weight initialization to suppress unnecessary duplicate frequencies, along with frequency selection as a practical solution. The experimental results demonstrate the effectiveness of the proposed approach.

**Strengths:**

This paper is well-written. It proposes that near-zero weight initialization can address the model’s frequency content and limitations. Furthermore, the paper introduces a frequency selection method to provide a practical solution.

**Weaknesses:**

The experimental results effectively illustrate the contributions, but I still have doubts about the practical applicability. How to extend the method to 2D should be clearly illustrated. Furthermore, regarding the dataset, a broader variety of data should be included in the experiments.

**Questions:**

see the weakness

---

> ### Author Response · Authors · 2025-11-18
> **2D near-zero weight initialization results added**
>
> Thank you very much for your comments, we were able to successfully apply the near-zero weight initialization technique now for the 2D case as well and have provided the results in section 5.3. Previously we had reduced the number of samples for the 2D experiments to 25 per dimension which was insufficient. With 50 samples per dim and a more expressive circuit, the near-zero init technique works now for the 2D experiments as well.

---

### Official Review · Reviewer_syP8 · 2025-11-02

**Soundness:** 3
**Presentation:** 3
**Contribution:** 2
**Rating:** 2
**Confidence:** 2

**Summary:**

The paper "Mitigating Exponential Mixed Frequency Growth", presents an analysis of parameter sufficiency conditions. The paper shows that duplicate frequencies are necessary for function approximation with quantum circuits. Furthermore the paper studies initialization, as well as parameter efficient training for quantum machine learning tasks.

The paper establishes minimum parameter requirements via the Frequency coverage condition,
Frequency control condition, and Optimization landscape conditions.

The paper presents quantum computer simulations, as experiments to back up its findings.

**Strengths:**

- Quantum machine learning can potentially unlock progress leaps in the future.

**Weaknesses:**

- The experimental section is does not reference or is integrated into related work.
- I am not sure it the approach to quantum machine learning presented here allows for non-linearity, which is common in the deep learning world.
- Section 5.1 (experimental setup) does not discuss the experimental setup at all. The supplementary finally states the we are looking at simulated quantum computing results from a consumer notebook. I am uncertain. How realistic are these simulations? Will they be useful in the future?

**Questions:**

- Is there a way to add non-linearity to quantum circuits?
- Is it possible to run some of these experiments on publicly available machines, i.e. the free tier of IBM-Q?
- According to the supplementary the optimization runs in Jax, to the best of my knowledge Jax would not run on a quantum computer. Would it even be possible to run this code on an actual machine instead of a simulator? How far is this work away from running on an actual device?

---

> ### Author Response · Authors · 2025-11-18
>
> Thanks a lot for your comments, especially for bringing up the question about running the models on actual quantum hardware. For a slightly different target function (with frequencies of 10, 20, 30) we had actually run the test set (with priviledged access) on IBM Fez, a 156-qubit Heron r2 processor with one of the lowest error rates currently available. As our training used a gradient-based optimizer and 5,000 iterations, we couldn't run it on the actual quantum hardware as it would have been too costly. We used however a noisy simulator (with the specifications for IBM Fez) and then subsequently only ran the test set on the actual quantum machine. The R2 Score obtained was 0.72 whereas the noiseless training and inference achieved 0.95. We chose not to repeat this experiment for the target function presented in the paper because our primary objective was to compare different techniques, which would have required running all variants on quantum hardware—a cost beyond our available budget.
> Jax was only used to speed up the simulation, the actual hardware runs didn't make use of Jax and simply relied on "standard" Pennylane.
>
> Thanks also for highlighting that the reference to the experimental setup section in Appendix A should have been included also in Section 5.1. We have now added it there as well.
>
> The non-linearity which proved very powerful in classical models is introduced into quantum models through (non-linear) rotational encoding gates, (conditional) 2-qubit gates and data-reuploading. Thanks once more for highlighting this important element, we have added this feature to the introduction (line 49f).

---

### Official Review · Reviewer_fHUG · 2025-11-07

**Soundness:** 1
**Presentation:** 2
**Contribution:** 1
**Rating:** 2
**Confidence:** 2

**Summary:**

This paper addresses the exponential growth of parameters in angle-encoded quantum models, which is exacerbated by mixed-frequency terms in high-dimensional data. Through white-box experiments, the authors identify a novel "additional parameter burden" stemming from the need to control a large number of non-unique (degenerate) frequencies. To mitigate this, they first propose a near-zero weight initialization heuristic, which proves effective for 1D problems by suppressing these unnecessary frequencies. Second, for cases where the target spectrum is known a priori, they introduce a "frequency selection" algorithm to build a sparse model spectrum that matches the target. This selection method is shown to achieve near-optimal performance in 2D examples while using significantly fewer parameters than standard dense-spectrum model

**Strengths:**

The paper's main strength is its focus on a significant and unresolved problem: the exponential parameter scaling in high-dimensional quantum models, which is a major bottleneck for the field. Its originality lies  in attempting to identify a specific experimental failure (the "serial unary" case) and proposing a new "additional parameter burden" from non-unique frequencies, which seeks to challenge existing theoretical conditions. The paper's clearest contribution is the experimental demonstration of the "frequency selection" algorithm, which shows that a model with a sparse spectrum can achieve high $R^2$ scores on 2D tasks with fewer parameters than dense models. The work is structured with reasonable clarity, providing a review of the theoretical background on Fourier analysis and DLA, and its potential for reproducibility is bolstered by an extensive appendix with detailed circuit diagrams. This "frequency selection" component, which demonstrates a practical engineering approach for sparse-spectrum targets, stands as the paper's most tangible result, even as its broader theoretical claims remain unsubstantiated

**Weaknesses:**

The paper's primary weakness is that its central theoretical claim—the "additional parameter burden" from non-unique frequencies—appears to be founded on a single, wrongly interpreted experiment. The "serial unary" model's failure is presented as evidence for a novel phenomenon, but this result can be explained by a direct violation of the known "frequency control condition" $(p\geq|Ω|)$, which the authors themselves cite. This experiment's 45 parameters are on a single qubit (Figure 10 in App. C) and are thus not linearly independent, a specific limitation the paper explicitly describes [lines 177-178] but fails to apply to its own key experiment.

This misinterpretation of a known failure mode as a new theoretical discovery significantly weakens the motivation for the proposed "near-zero weight initialization" heuristic. Furthermore, the paper's practical contributions are severely constrained by its own admissions: the "near-zero init" is a 1D-only solution that "could not be replicated for higher dimensional target functions".

Additionally the "frequency selection" algorithm is practically limited to toy problems, as it "exploits complete domain knowledge" of the target spectrum a priori.

**Questions:**

Your paper's most novel theoretical claim, the "additional parameter burden" from non-unique frequencies , rests almost entirely on the "unexpected case" of the 1D "serial unary" model's failure. You state this model, with 45 parameters, satisfies the known requirements $p \geq |Ω|$ (45 > 25) and $p \geq dim(g)$ (45 > 3). However, this is a single-qubit architecture (Figure 10). As you correctly state in Section 3, parameters added serially on the same qubit "fail to generate additional linearly independent Fourier coefficients".

Question: Can you provide a rigorous calculation or a detailed argument that the 45 parameters in the "serial unary" model are, in fact, linearly independent and that $p_{ind} \geq 25$? If they are not, isn't this model's failure an expected result of violating the known "frequency control condition", which would undermine the primary evidence for your "non-unique frequency" hypothesis?

The "near-zero weight initialization" heuristic is presented as a solution to the non-unique frequency burden, but you (commendably) note in Section 6 that these successful 1D results "could not be replicated for higher dimensional target functions". This lack of generalizability to 2D is a critical limitation for a paper addressing high-dimensional scaling.

Question: Can you provide any insight into why this heuristic fails to generalize? Does its failure in 2D suggest it is not addressing the hypothesized root cause (the non-unique frequency burden), but is perhaps an artifact of 1-qubit optimization dynamics?

The "frequency selection" algorithm is experimentally robust but is prefaced on "complete domain knowledge"  of the target function's frequency spectrum. You suggest this could be obtained via "classical Fourier analysis".

Question: For high-dimensional datasets (the paper's core motivation), performing a classical multi-dimensional Fourier transform is itself an exponentially hard, and thus intractable, problem. Could you elaborate on a realistic path to applying this method to high-dimensional, real-world data where the target spectrum is unknown and classically intractable to obtain?

---

> ### Author Response · Authors · 2025-11-18
>
> Thank you very much for pointing out the misleading statement in 177-178 about the linear independent parameters. The key requirement for being able to control frequencies separately is to be able to train coefficients for the frequencies to separate values. This cannot be done when parameter dependent gates are added consecutively on a single qubit as the additional parameters collapse together with the original parameters into just 3 parameters. When non-linearity is introduced however by data dependent rotational gates for the feature map between the ansatz layers, the parameters of the next ansatz layer can be trained to different values, leading to additional individual coefficients. We have been more specific now in line 175f and provide theoretical background on the model coefficients and frequencies in Appendix B.
> Apologies also for stating that the serial unary model has 45 parameters. This is wrong and stems from a previous version where we had used 15 ansatz layers. The correct number of individual parameters generated by 13 ansatz layers with 3 parameters each is 39 which has been updated in the latest version.
>
> We were able to successfully apply the near-zero weight initialization technique now for the 2D case as well and have provided the results in section 5.3. Previously we had reduced the number of samples for the 2D experiments to 25 per dimension which was insufficient. With 50 samples per dim and a more expressive circuit, the near-zero init technique works now for the 2D experiments as well.
>
> Thank you very much for pointing out that multi-dimensional Fourier analysis is infeasible. Our envisaged path to apply the frequency selection is to perform 1D Fourier analysis for each dimension separately and then allow the quantum model to combine these frequencies. With the help of trainable frequencies introduced by Jaderberg et al. we hope to be able to fit the unknown target function In future work even when the initial model 1D frequencies do not fully match the target frequencies.

---

### Official Review · Reviewer_7UiP · 2025-11-08

**Soundness:** 2
**Presentation:** 2
**Contribution:** 2
**Rating:** 4
**Confidence:** 4

**Summary:**

In the manuscript titled "Mitigating Exponential Mixed Frequency Growth Through Frequency Selection," the authors propose a practical approach to address the problem of exponential frequency growth in quantum machine learning. Their main contribution lies in a frequency selection strategy that significantly reduces the number of required parameters. The experimental design is sound, employing white-box target functions (Fourier series with known frequencies) for systematic evaluation, which effectively validates the efficacy of the proposed method. The frequency selection approach achieves comparable performance using only 78% of the parameters required by the best standard method. Finally, I have the following concerns and suggestions:

1. Section 5.3, "Near-Zero Weight Initialization," in the paper states that near-zero weight initialization works well in the 1D case but performs poorly in the 2D case. The authors do not explain why this method deteriorates in higher dimensions—could a more in-depth analysis be provided?
2. Section 3.1, "Multi-Dimensional Extensions and Mixed Frequencies," states that "every frequency in the spectrum requires individual coefficient control..." but fails to explain why, in quantum circuits, non-unique frequencies necessitate additional trainable parameters. In a classical Fourier series, each frequency component is independent—so why, in a quantum model, must extra parameters be allocated specifically to control non-unique (i.e., duplicate) frequencies? The authors are encouraged to supplement the manuscript with a theoretical analysis of the relationship between frequencies and parameters in quantum circuits, clarifying why controlling non-unique frequencies demands additional parameters.
3. The practicality of the method presented in the paper relies on prior knowledge of the target function's frequencies, which is often unrealistic in real-world scenarios. The paper does not discuss how to determine the frequencies of the target function in the absence of such prior knowledge. Could the authors provide clarification or suggestions on this aspect?
4. It is recommended to incorporate a preprocessing step—such as classical Fourier analysis (e.g., the method of Wiedmann et al., 2024, mentioned in the paper)—to extract frequencies, and to compare the performance gap and computational overhead between scenarios with and without prior frequency knowledge. This would help establish a logically complete picture of the method’s applicability.
5. In Section 4.3, "Frequency Selection," the statement "By selecting prefactors that deviate from the standard ternary..." lacks a clear description of the specific strategy for choosing these prefactors, providing only a single example with prefactors 3 and 9. The authors are encouraged to elaborate on this selection methodology.
In conclusion, before this paper is accepted for publication, the above-mentioned issues need to be addressed and revised.

**Strengths:**

The authors propose a practical approach to address the problem of exponential frequency growth in quantum machine learning. Their main contribution lies in a frequency selection strategy that significantly reduces the number of required parameters. The experimental design is sound, employing white-box target functions (Fourier series with known frequencies) for systematic evaluation, which effectively validates the efficacy of the proposed method. The frequency selection approach achieves comparable performance using only 78% of the parameters required by the best standard method.

**Weaknesses:**

1. Section 5.3, "Near-Zero Weight Initialization," in the paper states that near-zero weight initialization works well in the 1D case but performs poorly in the 2D case. The authors do not explain why this method deteriorates in higher dimensions—could a more in-depth analysis be provided?
2. Section 3.1, "Multi-Dimensional Extensions and Mixed Frequencies," states that "every frequency in the spectrum requires individual coefficient control..." but fails to explain why, in quantum circuits, non-unique frequencies necessitate additional trainable parameters. In a classical Fourier series, each frequency component is independent—so why, in a quantum model, must extra parameters be allocated specifically to control non-unique (i.e., duplicate) frequencies? The authors are encouraged to supplement the manuscript with a theoretical analysis of the relationship between frequencies and parameters in quantum circuits, clarifying why controlling non-unique frequencies demands additional parameters.
3. The practicality of the method presented in the paper relies on prior knowledge of the target function's frequencies, which is often unrealistic in real-world scenarios. The paper does not discuss how to determine the frequencies of the target function in the absence of such prior knowledge. Could the authors provide clarification or suggestions on this aspect?
4. It is recommended to incorporate a preprocessing step—such as classical Fourier analysis (e.g., the method of Wiedmann et al., 2024, mentioned in the paper)—to extract frequencies, and to compare the performance gap and computational overhead between scenarios with and without prior frequency knowledge. This would help establish a logically complete picture of the method’s applicability.
5. In Section 4.3, "Frequency Selection," the statement "By selecting prefactors that deviate from the standard ternary..." lacks a clear description of the specific strategy for choosing these prefactors, providing only a single example with prefactors 3 and 9. The authors are encouraged to elaborate on this selection methodology.

**Questions:**

1. Section 5.3, "Near-Zero Weight Initialization," in the paper states that near-zero weight initialization works well in the 1D case but performs poorly in the 2D case. The authors do not explain why this method deteriorates in higher dimensions—could a more in-depth analysis be provided?
2. Section 3.1, "Multi-Dimensional Extensions and Mixed Frequencies," states that "every frequency in the spectrum requires individual coefficient control..." but fails to explain why, in quantum circuits, non-unique frequencies necessitate additional trainable parameters. In a classical Fourier series, each frequency component is independent—so why, in a quantum model, must extra parameters be allocated specifically to control non-unique (i.e., duplicate) frequencies? The authors are encouraged to supplement the manuscript with a theoretical analysis of the relationship between frequencies and parameters in quantum circuits, clarifying why controlling non-unique frequencies demands additional parameters.
3. The practicality of the method presented in the paper relies on prior knowledge of the target function's frequencies, which is often unrealistic in real-world scenarios. The paper does not discuss how to determine the frequencies of the target function in the absence of such prior knowledge. Could the authors provide clarification or suggestions on this aspect?
4. It is recommended to incorporate a preprocessing step—such as classical Fourier analysis (e.g., the method of Wiedmann et al., 2024, mentioned in the paper)—to extract frequencies, and to compare the performance gap and computational overhead between scenarios with and without prior frequency knowledge. This would help establish a logically complete picture of the method’s applicability.
5. In Section 4.3, "Frequency Selection," the statement "By selecting prefactors that deviate from the standard ternary..." lacks a clear description of the specific strategy for choosing these prefactors, providing only a single example with prefactors 3 and 9. The authors are encouraged to elaborate on this selection methodology.

---

> ### Author Response · Authors · 2025-11-18
>
> Thank you very much for your valuable comments:
> 1. We were able to successfully apply the near-zero weight initialization technique now for the 2D case as well and have provided the results in section 5.3. Previously we had reduced the number of samples for the 2D experiments to 25 per dimension which was insufficient. With 50 samples per dim and a more expressive circuit, the near-zero init technique works now for the 2D experiments as well.
> 2. Thank you very much for highlighting this essential necessity: the multiple model coefficients and frequencies need to match the unique target coefficients and frequencies. In Appendix B we have added a detailed explanation how such multiple model coefficients and frequencies are generated and which challenges this creates.
> 3. to  5. For real-world datasets we envisage the following path in order to apply the frequency selection technique. i) perform 1D Fourier analysis to obtain the most prominent frequencies in each dimension, ii) use combinatorial optimization to identify the prefactors that can provide these frequencies (a naive method would be to use prefactors of 1, f1+1, f2+1, etc. for frequencies of f1, f2, etc. since the frequencies are generated by the differences in the eigenvectors of the generator Hamiltonians). As long as this results in less FMs than the ternary FMs (which densely cover the full frequency spectrum) with prefactors larger than the powers of 3, this will create gaps in the frequency spectrum and consequently generate less total frequencies that need to be controlled. In future work, we aim to perform this on a real-world data set.

---

### Meta-Review · Area_Chair_mA3x · 2025-12-23

**Summary:**

This paper studies the exponential growth of mixed Fourier frequencies in angle-encoded quantum machine learning models and their impact on trainability. Through controlled white-box experiments with known target spectra, the authors identify an additional parameter burden arising from non-unique (duplicate) frequencies in multi-dimensional settings. To mitigate this, they propose two strategies: near-zero weight initialization to suppress unnecessary frequencies, and frequency selection when target frequencies are known a priori, which constructs a sparse model spectrum. Experiments show that frequency selection can achieve near-optimal performance in low-dimensional tasks while using significantly fewer parameters than standard dense-spectrum approaches.

Reviewer opinions were largely borderline to negative. Several serious concerns were raised. Most notably, reviewers identified factual and conceptual errors in the theoretical interpretation of key experiments, which the authors later acknowledged and corrected during the rebuttal. This raised concerns about the overall rigor and reliability of the analysis. Additional issues included the heavy reliance on prior knowledge of target frequencies, the limited scope of experiments restricted to synthetic, low-dimensional settings, and the unclear practical relevance of the proposed frequency selection strategy for real-world learning tasks. Reviewers also questioned whether the claimed “additional parameter burden” represents a genuinely new phenomenon beyond known limitations of frequency control. Given these issues, I recommend rejection.

**Reviewer Concerns:**

**Concerns Addressed by the Rebuttal**

The rebuttal addressed some technical and experimental clarification issues. In particular, the authors provided additional experimental results for the 2D setting of near-zero weight initialization and clarified several implementation details, including how non-unique frequencies arise from parameterized quantum circuits and the source of nonlinearity in the model. Some ambiguities in experimental descriptions were also clarified.

**Concerns Remaining After the Rebuttal**

Several major concerns remain unresolved. Most importantly, the reviewers identified factual errors and theoretical misinterpretations in the core argument (e.g., regarding parameter independence and the interpretation of key experiments), which the authors acknowledged and corrected in the rebuttal. This goes beyond a presentation issue and significantly undermines the reliability of the central theoretical claims.

Moreover, the main claim of an “additional parameter burden” caused by non-unique frequencies is still not convincingly distinguished from known limitations of frequency control in angle-encoded models. The proposed frequency selection strategy relies heavily on prior knowledge of target spectra and is supported only by limited, white-box experiments, leaving its practicality for real-world tasks unsubstantiated. Finally, despite some additional experiments, the empirical evidence remains restricted to low-dimensional, highly controlled settings, limiting the generality and impact of the conclusions.

**Reviewer Scores:**

Given that the rebuttal acknowledged and corrected several factual errors but did not resolve the core theoretical and practical concerns raised by the reviewers, it is unlikely that any reviewer would have increased their score had they participated fully in the discussion. At best, the scores would have remained unchanged, and for some reviewers, the identified issues could have further reinforced their initial assessments.

---

### Decision · Program_Chairs · 2026-01-26

Reject